# Air Pollution in Kosovo: Short Term Effects on Hospital Visits of Children Due to Respiratory Health Diagnoses

**DOI:** 10.3390/ijerph191610141

**Published:** 2022-08-16

**Authors:** Zana Shabani Isenaj, Merita Berisha, Dragan Gjorgjev, Mirjana Dimovska, Hanns Moshammer, Antigona Ukëhaxhaj

**Affiliations:** 1Medical Faculty, University of Hasan Pristina, George Bush 31, 10000 Pristina, Kosovo; 2National Institute of Public Health, St. Mother Teresa pn, Rrethi i Hospitalit, 10000 Pristina, Kosovo; 3Faculty of Medicine, Ss. Cyril and Methodius University in Skopje, 1000 Skopje, North Macedonia; 4Department of Environmental Health, ZPH, Medical University of Vienna, 1090 Vienna, Austria; 5Department of Hygiene, Medical University of Karakalpakstan, Nukus 230100, Uzbekistan; 6Faculty of Medicine, University Fehmi Agani, Ismail Qemali n.n., 50000 Gjakove, Kosovo

**Keywords:** PM2.5, hospital admissions, ambulatory visits, time series analysis, lag effects

## Abstract

The Republic of Kosovo is a small country in the Balkans. The capital city of Pristina hosts most of its population and is situated in a mountain basin with poor air exchange, especially during winter. Domestic heating, road transport, industry and coal-fired power plants contribute to high levels of air pollution. We performed a time-series analysis on effects of particulate air pollution (PM2.5) on respiratory health of children and adolescents, using hospital admission and ambulatory visit numbers from the pediatric university clinic. From 2018 until 2020, daily mean concentrations of PM2.5 ranged between 2.41 and 161.03 µg/m³. On average, there were 6.7 ambulatory visits per day with lower numbers on weekends and during the first COVID-19 wave in 2020. An increase in PM2.5 led to an immediate increase in visit numbers that lasted over several days. Averaged over a full week, this amounted to about a 1% increase per 10 µg/m³. There were, on average, 1.7 hospital admissions per day. Two and three days after a rise in air pollution, there was also a rise in admission numbers, followed by a decline during the consecutive days. This might indicate that the wards were overstressed because of high admission numbers and restricted additional admissions.

## 1. Introduction

Air pollution is the largest contributor to the environmental burden of disease, causing and exacerbating several diseases including cardiovascular diseases such as stroke and ischemic cardiopathy, acute respiratory infections, chronic obstructive pulmonary disease and cancer [1]. According to the World Health Organization (WHO), in 2012, one out of every nine deaths was the result of air pollution-related conditions. Of these deaths, around 3 million are attributable solely to ambient (outdoor) air pollution [2]. Respiratory diseases are among the leading causes of both morbidity and mortality worldwide. Lung infections, chronic obstructive pulmonary diseases, asthma, and bronchial cancer are main contributors of respiratory-related mortality and morbidity [3,4]. By 2030, the WHO estimates that respiratory diseases will be the most important reason for impaired quality of life and death within the EU that will require substantial economic resources for both prevention and treatment [5].

The strongest and most consistent association between air pollution exposure and respiratory morbidities has been seen for ambient particulate matter. Particulate matter (PM) is a mixture of solid particles and liquid droplets that vary in size and origin. Commonly used indicators describing PM that are relevant to health refer to the mass concentration of particles with a diameter of less than 10 µm (PM10) and of particles with a diameter of less than 2.5 µm (PM2.5). Findings from many epidemiological and toxicological studies have highlighted adverse effects of air pollution on both premature mortality and morbidity from respiratory and cardiovascular disease, following both short-term and chronic exposure [6,7]. Apart from other vulnerable groups, children are also susceptible to air pollution. For example, we have shown short-term effects of air pollution on lung function in asthmatic children [8]. We also demonstrated long-term beneficial effects on lung function with declining air pollution [9]. The relationship between air pollutants and respiratory hospital admissions has been reported both in developed countries and in developing countries [10,11]. Studies have also demonstrated a significant impact of air pollution on infants and children, which is manifested primarily as a range of respiratory problems [12].

Several epidemiological studies have found associations between daily concentrations of ambient air pollution, such as particulate matter (PM), ozone (O_3_), and nitrogen dioxide (NO_2_), and daily mortality and morbidity outcomes due to respiratory diseases [13,14,15,16,17]. Children are particularly susceptible to ambient air pollution due to their physiological and behavioral characteristics. Previous studies found that children were the most susceptible to hospital admission for respiratory disease associated with PM levels and meteorological factors [18,19,20,21]. Children are considered to be more sensitive to air pollution than adults, and children with asthma are particularly vulnerable to the adverse health effects of air pollution [22]. Ségala and colleagues [23] found PM10 and nitrogen dioxide (NO_2_) to be positively associated with consultations and hospitalizations for bronchiolitis during the winter. Finally, Rahman and colleagues [24] found higher odds of acute bronchiolitis associated with exposure to PM10, among children who reside in urban areas. A recent study revealed that acute bronchiolitis-related hospitalization among children was associated with temperature and exposure to NO_2_ and PM10 [25] suggesting a need to adopt sustainable clean air policies to protect young children’s health.

A significant proportion of urban areas in Kosovo suffer from poor air quality, with ambient concentrations of particulate matter with a diameter of 2.5 μm or less (PM2.5) significantly exceeding the national and European Union (EU) standards (20 µg/m³ annual mean [26]) and global air quality guidelines for PM2.5 (5 µg/m³ annual mean [27]) established by the WHO [28]. Air pollution is mainly associated with the burning of solid fuels in homes, obsolete large thermal power plants, industry and exhaust gas from vehicles and has become a serious environmental problem, and its effect on human health is a concern [29].

Kosovo has one of the youngest populations in Europe accompanied by a low life expectancy of 71.6 years, the lowest in the region [30]. The population in Kosovo is growing with now reaching about 1.8 million inhabitants, more than 650,000 of which are aged 19 or younger [31]. The health care system is composed of the public health care network (primary, secondary and tertiary levels) and facilities in private ownership, funded largely from government and municipal tax and direct payments [32]. The main concerns in relation to the health system are the quality of health services and low financing. A 2014 report stated that citizens in Kosovo had to cover expenses for health services with out-of-pocket expenditures ranging from 30 to 50% of total costs [31]. At the same time, the launch of ambitious healthcare reforms, in combination with the fragile structure and governance of the system, have impacted the overall stability and continuity of the service provision [32].

Although the air pollution burden in Kosovo is not well-documented, in part due to not fully functional health information system, a recent report from the World Bank estimates that about 760 people die prematurely every year in Kosovo because of exposure to ambient air pollution, where 11 percent of these cases are from Pristina, representing the most polluted and most populated city (more than 200,000 persons in 2021 [31]) in Kosovo. Of the total number of pollution-related deaths, 90 percent are due to ischemic heart disease and stroke combined [28]. The capital city, Pristina, faces severe smog episodes in the winter, rivaling those of big cities such as Beijing, Mumbai and New Delhi–causing significant health impact [28]. This study aims to examine associations between air pollution and children’s health, thus contributing to local data-based evidence of the environmental burden of disease in Kosovo.

## 2. Materials and Methods

### 2.1. Air Pollution Data

Kosovo is a small land-locked country located in the mountainous regions of the Balkan Range. Most of its population lives in the shallow basin in and around the capital city of Pristina and most people living in this area most likely experience a similar time-course of air pollution exposure. The whole population in this basin is served by the hospitals in Pristina with the pediatric university clinic being the primary point of care for children and adolescents. 

Data from the monitoring network are collected by the Kosovo Hydro-Meteorological Institute (KHMI) and reported to the European Environment Agency (EEA) [33] and published on the KHMI website [34]. Daily averages were available from both sources starting from 13 May 2018. Over time, the number of monitoring stations and completeness of data increased for all of Kosovo. Daily data for PM2.5, PM10, and NO_2_, were collected from the EEA website from all available stations until 1 March 2022. Daily average temperature data from Pristina (station at KHMI) were available in annual meteorological reports. The latest report available at the time of data collection early in 2022 was the 2020 report. The US embassy in Pristina measures PM2.5 since several years and daily and hourly values are reported on the AirNow website of the US government [35]. Daily data were extracted from 23 March 2016, until 1 March 2022.

### 2.2. Health Data

The number of daily ambulatory visits from the district of Pristina were obtained from the Health Information Statistics Platform from Ministry of Health in Kosovo, and also retrieved manually from the protocols of specialist ambulatories of the Pediatric Clinic at the University Clinical Center of Kosovo. Data of the daily hospital admissions were obtained manually from the protocols of the Pediatric Clinic from the University Clinical Center of Kosovo, due to the lack of electronic data. The data included basic information such as sex, age, date of visit or hospital admission and disease diagnosis.

We included data from all children aged 0–18 years admitted to the hospital with respiratory disease (International Classification of Diseases, 10th Revision: J00-J99) between January 2018 and December 2020 and children who visited the ambulatories. Patients who came from locations other than municipality of Pristina were excluded.

### 2.3. Statistical Analysis

In a first preparatory step, the pairwise Pearson correlation between monitoring stations was assessed for all pollutants and both for the whole observation period and also stratified by month. The pollutant with the strongest correlation between stations was selected as the exposure variable. Daily data were primarily taken from the station that was most closely correlated with the other stations. Missing data at that station were replaced by estimated values based on linear regression between the index station and the other station that was most closely correlated with it.

Next, for each of the two outcomes (number of ambulatory visits and number of hospital admissions), a separate General Additive Model (GAM) was constructed, using the library mgcv in R [36]. To that end, a script was adapted that was originally written by Daniel Rabczenko for the analysis of daily deaths in Vienna, Graz and Linz [37,38]. That script used over-dispersed Poisson GAMs [39], applying a 7-day polynomially distributed lag (pdl) model [40] and stringent convergence criteria [41,42], considering day of the week as “as factor” variable. Seasonal and longer-term variations were modelled using spline functions [43]. In order to find the optimal number of knots (or degrees of freedom) for the non-linear fit, we chose the model with the smallest partial autocorrelation (sum over 30 days). Next, we added temperature (natural spline) to the model. We selected the most relevant lag (lags 0–3) based on the Akaike information criterion (AIC) [44]. Next, we performed the same selection process on the change in temperature between consecutive days. As a sensitivity analysis, splines for the temporal variation were also examined with different number of knots. In addition, effects of single lags were also assessed to check if a third-degree polynomial was a valid approximation of the lag structure of the PM2.5 impact.

## 3. Results

### 3.1. Air Pollution Data

PM2.5 displayed the strongest temporal correlation between monitoring stations. The data from the KHMI urban background station were highly correlated with all other stations situated in urban or sub-urban backgrounds. Pearson’s correlation coefficient was generally larger than 0.8, with the exception of the urban background station of Prizren, for which the correlation coefficient with the KHMI station was only 0.765. Details of the monitoring stations in Kosovo and the temporal correlation coefficients between the stations is provided in Appendix A. PM10 and NO_2_ displayed somewhat weaker correlations, although still quite strong and usually stronger in winter. PM2.5 was chosen as the best measure of exposure, because its time-series reflected best the temporal exposure variation of the total population. Moreover, for this pollutant the US embassy provided an additional series that was strongly correlated with the KHMI urban background station (r = 0.964) that allowed for filling the gaps on days with missing KHMI data using the linear regression formula:KHMI = 0.8165507 × embassy + 1.152676(1)

The average concentration of PM2.5 was 24.14 µg/m³ and the first, second (median) and third quartile were 11.14, 16.01 and 28.35 µg/m³ (IQR = 12.34). The maximal and the minimal daily concentrations were 2.41 and 161.03 µg/m³.

### 3.2. Time Course of Ambulatory Visits and Hospital Admissions

Based on the above inclusion and exclusion criteria, a total of 1838 hospital admission records and 7372 ambulatory visits were collected during this study period. On average, there were 6.7 ambulatory visits per day (quartiles: 0, 4, 12, maximum 96). On average, 1.7 children were admitted per day because of respiratory diseases (quartiles: 0, 1, 3, maximum: 18).

Both time-series of health data displayed some seasonal fluctuation with somewhat higher numbers in winter. However, most striking was a steep decline both in ambulatory visits and hospital admissions during the first COVID-19 wave in 2020 (Figure 1). Moreover, during weekends much fewer events occurred.

### 3.3. Short-Term Effect of Particulate Air Pollution (PM2.5)

An increase in air pollution led to an immediate (same day) increase in ambulatory visits. This increase was significant on the first and second day after the pollution episode. Afterwards, there was a decline in visits. However, added over a whole week, the total effect of an increase in air pollution was significantly positive (Figure 2). An increase in PM2.5 daily mean of 10 µg/m³, added over a full week, resulted in an increase in hospital admissions by 0.99% (95% confidence interval: 0.4; 1.6). 

Stationary hospital admissions only increased on the day following the pollution episode This effect was significant on the second and third day after the pollution episode and afterwards declined again and finally turned negative (Figure 2). Summed over the whole week after the pollution episode, no significant effect on hospital admissions could be observed (−0.2%; CI: −2.6; 2.3%).

For both outcomes, changing the number of knots for the temporal/seasonal variation did not substantially affect the main results (Figure A1, Figure A2 and Figure A3). Single lag analyses confirmed that a third-degree polynomial provided a reasonable approximation of the true lag structure (Figure A4).

## 4. Discussion

A steep decline in the number of ambulatory visits and hospital admissions was observed during the first wave of COVID-19. This phenomenon was attributed to the decrease in the demand for services from the citizens due to the restrictive measures introduced to mitigate the COVID-19 outbreak and the repurposing of health services and health workforce capacities to COVID-19 treatment centers. According to a report from WHO, this situation was also impacted by the lack of Essential Health Service Packages and lack of additional funds to maintain EHS. A rapid assessment of the situation analysis of the impact of COVID-19 in EHS revealed that all non-acute health services in Kosovo were suspended from March till May 2020, with a gradual re-opening afterwards [45]. In addition, individual protective measures such as social distancing and wearing of face masks could also have prevented other respiratory diseases.

This study is only based on three years of observation. This is not very long for a time series analysis. The study may therefore lack sufficient power to detect small effects. However, a longer time series was not feasible due to limitations in data availability. Temperature data (a relevant confounder) were only available until 2020. Air pollution data only became available in 2018. However, our findings are plausible. 

We did see an increase in ambulatory visits with increasing levels of PM2.5. This effect was already observed on the same day but reached significance only on the first and second day after the pollution episode. Averaged over 1 week, the effect size per 10 µg/µ³ was about 1%. This order of magnitude has been observed in other similar studies [46,47,48,49]. 

We expected to find an increase in case numbers soon after the pollution day and a decline in numbers a few days later. Such a shape is well represented by a third-degree polynomial and therefore, we decided in advance to apply such a function in the distributed lag model. Before applying the polynomial, we calculated the effect estimates for single lags to make sure that our assumption was correct. Such a single lag estimate is confounded by the effects from neighboring lags because of strong short-term autocorrelation of pollution concentrations. Therefore, single-day lags cannot inform about the true effect strength but provide valid information about the overall temporal distribution of the effect.

This bi-phasic shape of the temporal exposure-response distribution has been discussed for several decades under the term “harvesting hypothesis” or “harvesting effect” [50,51,52,53,54,55]. Maybe, the term “harvesting” originally really referred to the work of a farmer: During harvest time the farmer will try to harvest the crops during sunny weather, because when the harvest is performed in rainy weather, there is an increased danger of mold. Therefore, in a time series, sunny days will be associated with a higher harvest yield. However, after a sunny day, all the crops on the farmer’s field will already have been harvested. So, on a later day, independent from rain or sunshine, a lower yield is to be expected. However, when this analogy was applied to time-series studies on daily mortality, the term “harvesting” quickly got associated with the mythical concept of the grim reaper. Indeed, some authors argued that an increase in death numbers at or soon after a high pollution day was not a sign of a relevant health effect because death only affected those that otherwise would have died just a few days later. However, further analyses (also applying distributed lag models [52] or compartment models [54]) confirmed, that only a small part of the excess mortality is only due to very short term replacement (“harvesting”). 

Of course, every person can only die once. Therefore, when more people out of the “very frail pool” [53] die on one day, there are less people in this pool left to die on a consecutive day. With hospital admissions and ambulatory visits, this is different in the sense that patients can also count as “cases” more often than once. Still, an increase in visit frequency per person would also be a relevant effect of air pollution. Due to anonymous data, we cannot check how many of the visits in our study are first or repeated visits. However, this does not invalidate the results of our study. In comparison, although less well studied, a biphasic temporal shape of the effect estimate is also expected for visits. 

Thus, relative risks obtained in time-series analyses refer to the vulnerable group mostly and cannot be applied directly to the general population. Therefore, an estimate of absolute effects is provided here: Per week, there were on average 47 ambulatory visits. An increase per 10 µg/m³ on one day, would therefore lead to about one-half additional visit (0.47%) spread over the coming week. Considering that 10 µg/m³ are nearly the interquartile range of 12.34, achieving a reduction in PM2.5 by 10 µg/m³ would certainly not be an easy task. However, over the whole year, this would reduce the number of visits by about ½ per day, or by about 180 visits. Since the annual mean concentration of PM2.5 was even more than twice as high (24.14 µg/m³), PM2.5 in total was responsible for approximately 1 visit per day. Moreover, this is likely an underestimation, given the fact that air pollution concentrations peak in winter when hospital admissions are highest for respiratory diseases. It is also an underestimation, because part of the study coincided with the 2020 corona lock-down measures. Of course, this is only considering one single outcome (respiratory diseases) and only in the child and adolescent population. Additionally, this impact only considers one although the most important hospital serving this population in Kosovo.

Stationary hospital admissions rose after an increase in air pollution with a latency of one day. This seems plausible, as admissions usually need a consultation with a general practitioner/a pediatrician in primary health care first. This step would allow for a delay of one day. In Austria, we even observed a longer delay for hospital visits because of respiratory diagnoses (2 days for city dwellers and even longer for rural people) [56]. 

Most likely because of this delay, the biphasic curve produced by the polynomial distributed lag model had not returned to the baseline by lag 6. Therefore, we also examined a longer lag period of 10 days (lag 0–lag 9, Figure A3). In that model, the biphasic curve is clearly complete, but the point estimates for the later days are really imprecise. Data from only 3 years seem insufficient for the analysis of longer lag periods. Thus, we believe that a lag period of 7 days was a good choice. 

After this delay, for stationary admissions a significant effect of air pollution was also seen. However, this increase was followed by a decrease on days 5 and 6. Thus, over the whole week, air pollution did not lead to an increase in admissions. If this was to be interpreted as a “perfect harvesting effect” whereby “all crops that could be harvested, had been harvested on the first sunny day”, it would mean that “all children that could be admitted to the hospital have already been admitted during the first days of the pollution episode”. This interpretation is unlikely. It more likely demonstrates that “all hospital beds that could be occupied were already occupied soon after the beginning of each pollution episode, indicating that the hospital ward is already reaching the limit of its resources. When the ward gets overfilled, either due to more admissions or longer duration of stay because of more severe cases, the doctors will be more restrictive in accepting new patients. Therefore, a drop in admissions after the peak triggered by air pollution very likely indicates that the hospital resources are strained to their limits by an air pollution episode. Moreover, in this respect it is noteworthy, that air pollution episodes occur mostly in wintertime, when respiratory diseases are already most frequent. Interpreted this way, a net zero effect over a week in spite of a clear and plausible response on single days, should rather be a cause for concern.

## 5. Conclusions

The study found plausible and consistent short-term effects of particulate air pollution on respiratory morbidity of children. The effect was most clearly seen in the number of ambulatory visits. It is generally established that short-term effects are not as important as the long-term health consequences of chronic air pollution exposure. However, the study of chronic effects is much more difficult, as it requires longer series of monitoring data. Time-series analyses are an elegant way of demonstrating health effects of air pollution. However, impact assessments solely based on the results of time-series analyses would severely underestimate the magnitude of the impact. Nevertheless, even this time-series study clearly demonstrates the dire need for air pollution abatement measures in Kosovo. 

## Figures and Tables

**Figure 1 ijerph-19-10141-f001:**
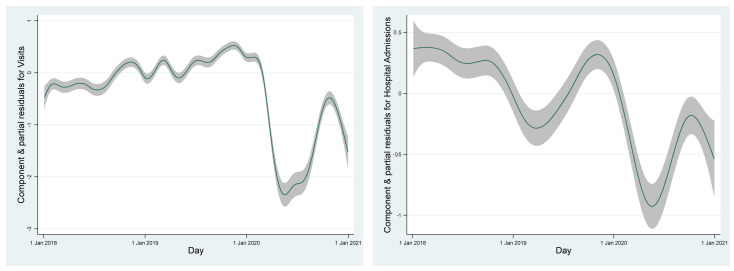
Smoothed time course of daily ambulatory visits (**left**) and stationary hospital admissions (**right**).

**Figure 2 ijerph-19-10141-f002:**
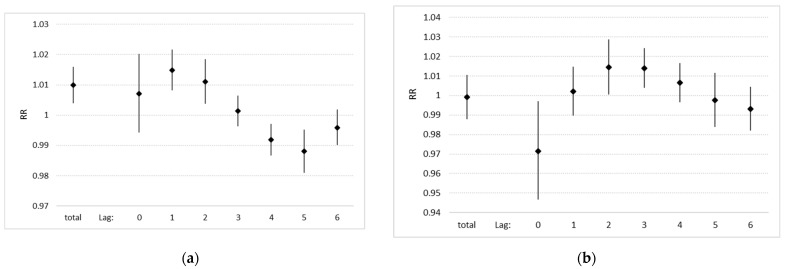
Effect (relative risk and 95% confidence interval) per 10 µg/m³ increase in PM2.5, total effect and daily effects over 7 consecutive days (lag 0–6): (**a**) ambulatory visits, (**b**) stationary hospital admissions.

## Data Availability

Air pollution and temperature data have been obtained from the internet from freely available sources. The links are provided in the text. Numbers of ambulatory visits and of stationary hospital admissions are available upon request. Send an e-mail to the corresponding author.

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
