# Peer review of "Air Pollution in Kosovo: Short Term Effects on Hospital Visits of Children Due to Respiratory Health Diagnoses"

_ijerph, 2022, doi:10.3390/ijerph191610141_

Round 1

Reviewer 1 Report

This article investigated the correlations between particulate air pollution and respiratory disease occurrence of children in Kosovo by performing a time-series analysis based on hospital admission and ambulatory visit numbers from the pediatric university clinic. The structure of the article is clear and sound. The discussion and conclusions are supported by the analysis in the results. Therefore, publication of this article on International Journal of Environmental Research and Public Health is recommended. A few comments are listed below. 

1. The title stated, "particulate air pollution", whereas this study also investigated the effect of NO2. It may be more accurate to avoid using "particulate" in the title.

2. In Abstract, it mentioned that they performed a time-series analysis using hospital admission and ambulatory visit numbers. One suggestion is that, based on the statements in Method section, it would be more accurate to clarify that only the respiratory disease related cases were analyzed in the research.

3. Lines 53-55, please provide proper references for this statement.

4. First sentence of the second last paragraph (lines 84-86), what does this statement indicate? The death rate is related to many aspects and complicated for analysis. Please provide more information if the authors would like to discuss the death rate and age distribution in Kosovo.

5.  Regarding the Figure 2, does the y-axis have a label? I am no expert in data science/analysis. Just out of curiosity.

Author Response

please see attached word document

Reviewer 2 Report

This is an observational study in Kosovo that aims to examine relationships between ambient particulate air pollution and hospitalization statistics in short time scales using daily data in 2018-2020. Short-term exposure to intense air pollution can be a serious health issue for vulnerable young populations, especially for modernized cities in developing countries. The study could also be interesting given the COVID phases in between. Using ambient monitoring data, the author determined PM2.5 as the best measure of exposure for Kosovo populations, computed their effect sizes for both ambulatory visits and stationary hospitalizations through lag analysis, and found increases in ambulatory visits with increasing PM2.5 levels but only statistically significant in the first two days in each pollution episode. However, the title of the manuscript is rather ambitious including “respiratory health of children”, and several drawbacks need to be addressed.

Major comments

1. Title. “respiratory health of children” is an overstatement; tone must be tuned down. Maybe change the whole title to “Particulate air pollution in Kosovo: short term effects on hospital visits of children with respiratory issues” or similar texts.

2. Abstract. Implications and Conclusion from these data are not adequately described within the grand theme of the manuscript, and no measures are discussed to further address potential health/healthcare problems spotted. The part of overstressed ward was untoward, deflecting, and bear little scientific advancement linking short-term air pollution to children’s respiratory health.

3. Health data - the author used ambulance visits and hospital admissions as measures of children’s respiratory health, which can be a biased and inprecise metric for the general population:

Are these visits representative of the general juvenile population in Kosovo? How large were these visits/admissions accounted for in the overall prevalence of respiratory issues?

Were there returning visits from the same subjects? The demographic data should be shown.

How did you control for confounders such as COVID-driven repurposing of healthcare system, lockdown measures, and likely underlying conditions of subjects such as asthma that either affect the actual admission numbers or proper inclusion of subjects in the study?

How did you differentiate respiratory problems caused by COVID and those potentially due to ambient pollutions? Are COVID diagnostics available?

5. Discussion. This is the section that should engage with relevant scientific field with most citations. The few cited literatures are only related to air pollution data and no references were cited for short-term effects of air pollution on children's health - no actual scientific discourse involved here.

Minor comments

Line 196. Define and explain “harvesting” effect.

Line 207. “Data not shown”. Please include the results with different knots in Appendix.

Author Response

please see attached word document

Reviewer 3 Report

Overall this is a very interesting analysis that adds to the body of literature on the severe health effects of PM 2.5.  Several issues warrant further explanation, including the conclusion about hospital admission restriction. Additionally, there are sentences that read awkwardly in English. The conclusion that the decrease if hospital admissions at the end of the week is due to people being turned away from hospitals needs to be supported by data, which should be obtainable.   

56 -  Citation?

76 – Excessive comma

78 – What are the standards?  Having the number would allow for a comparison to the concentrations measured in Kosovo.

86 – ‘Health Care’ doesn’t need to be capitalized unless it is a proper noun

91- ‘packet’ should be ‘pocket’

87-93 – 2014 was 8 years ago, yet reforms are discussed.  Are citizens still paying for health care?  What type of reforms?  How has the governance of the system impacted delivery?

96 – If the health information system is not fully functional, how reliable was hospital data used in this analysis?

95-104 – It would be helpful to know the population of Kosovo and the Prishtina, particularly as absolute measures for health conditions are given. 

107-123 – A map of the area, with the discussed topography and the location of the air monitors and the hospital would be helpful.

180-189 – What is the total population of children?  Again, absolute numbers are hard to interpret without context.

186 – Did PM 2.5 concentrations decrease with quarantine measures?  

196 – Define ‘harvesting’ effect.

237-8 – The results are presented as 1% (½ visit increase per week) per 10 ug/m3.  47 visits per week is 2444 visits per year.  A 1% reduction is 24.4 visits.  How did this turn to ½ visit per day?  Please clarify.

253-259 –Hospital admissions decreased at the end of the week – did PM 2.5 concentrations stay constant, increase, or decrease?  If the concentrations were constant, it is possible that the children affected would already be in the hospital?  What is the hospital bed capacity?  Does admittance rate exceed this? The conclusion that this decrease is due to people being turned away from hospitals needs to be supported by data, which should be obtainable.   

Author Response

please see attached word document

Round 2

Reviewer 2 Report

Although throughout the rebuttal, the authors have serious attitudinal problems that are disrespectful, unprofessional and unproductive, overall, with a new title, the revisions have addressed most of my concern. The revised manuscript is thus recommended for publication in the current form in IJERPH.

1. Title. The new title fits better with the scope / content of their study.

2. Abstract. Wasn’t sure if the authors have read my comments carefully – it was NOT the “measures against air pollution” I discussed, I was speaking of “measures against health/healthcare problems spotted”. I mentioned this because this was actually much highlighted in their own original title. If the authors did put “respiratory health of children” in the Title, it is then their responsibility to address those in the main texts and mention in the Abstract.

The overstressed ward part is an interesting finding; since the title has been modified, it now fits in better.

3-4 Health data

The justifications make much more sense. Please be sure to include these in manuscript as well. I suggest the authors make specific Line # of revised texts explicitly when addressing each point of concern / question, rather than claiming “we will discuss..”

5. Discussion

Addressed.

Minor comments

Addressed.